# Optimum Parameters for Extracting Three Kinds of Carotenoids from Pepper Leaves by Response Surface Methodology

**Nenghui Li [†], Jing Li [†], Dongxia Ding, Jianming Xie \*, Jing Zhang, Wangxiong Li, Yufeng Ma, Feng Gao, Tianhang Niu, Cheng Wang and Emily Patience Bakpa**

College of Horticulture, Gansu Agricultural University, Yingmen Village, Anning District, Lanzhou 730070, China; Linh@st.gsau.edu.cn (N.L.); lj@gsau.edu.cn (J.L.); dingdongxia100741@outlook.com (D.D.); zhangjing2019@yeah.net (J.Z.); Lwx0821@outlook.com (W.L.); mayf@st.gsau.edu.cn (Y.M.); g295641465@gmail.com (F.G.); Niutianhang@outlook.com (T.N.); gsauphd0810@outlook.com (C.W.); emilybakpa@gmail.com (E.P.B.)
* Correspondence: xiejianming@126.com; Tel.: +86-13893335780
† These authors contributed equally to this work.

**Abstract:** To determine the optimum parameters for extracting three carotenoids including zeaxanthin, lutein epoxide, and violaxanthin from pepper leaves by response surface methodology (RSM), a solvent of acetone and ethyl acetate (1:2) was used to extract carotenoids with four independent factors: ultrasound time (20–60 min); ratio of sample to solvent (1:12–1:4); saponification time (10–50 min); and concentration of saponification solution (KOH–methanol) (10–30%). A second-order polynomial model produced a satisfactory fitting of the experimental data with regard to zeaxanthin ($R^2$ = 75.95%, $p < 0.0197$), lutein epoxide ($R^2$ = 90.24%, $p < 0.0001$), and violaxanthin ($R^2$ = 73.84%, $p < 0.0809$) content. The optimum joint extraction conditions of zeaxanthin, lutein epoxide, and violaxanthin were 40 min, 1:8, 32 min, and 20%, respectively. The optimal predicted contents for zeaxanthin (0.823022 µg/g DW), lutein epoxide (4.03684 µg/g dry; DW—dry weight), and violaxanthin (16.1972 µg/g DW) in extraction had little difference with the actual experimental values obtained under the optimum extraction conditions for each response: zeaxanthin (0.8118 µg/g DW), lutein epoxide (3.9497 µg/g DW), and violaxanthin (16.1590 µg/g DW), which provides a theoretical basis and method for cultivating new varieties at low temperatures and weak light resistance.

**Keywords:** pepper leaves; carotenoids; response surface methodology; high performance liquid chromatography (HPLC); optimum



## 1. Introduction

Pepper (*Solanaceae: Capsicum annuum* L.) is the most vital off-season vegetable cultivated in greenhouses in northwestern China [1,2]. As one of the most favored vegetables, with its fruit mainly valued as a food seasoning, pepper fruit is also an excellent source of natural pigments, including neoflavin, cyanin, monoepoxy zeaxanthin, lutein, zeaxanthin, lutein epoxide, lycopene, octet lycopene, α-carotene, and β-carotene [3], which are responsible for the fruit's color that ranges from yellow, to orange, to red [4]. However, carotenoids in pepper leaves, associated with the tolerance mechanisms to low temperatures and low light levels, have not been studied extensively.

Carotenoids, a large group of natural pigments in animals, bacteria, and plants, are important ingredients in clinical and healthy foods [5–7]. For example, lutein and zeaxanthin, with strong antioxidant activity, substantially relieve visual fatigue and reduce the risk of macular degeneration and cataracts [8]. Numerous studies on carotenoids are mainly in relation to food science and chemistry. Most of these studies focused on changes in the concentrations and quantities of carotenoids in fruits or vegetables, such as red pepper [3], citrus [9], durian [10], red navel orange [11], goldenberry [12], and mango [13] during maturity, storage, and processing cycles. Carotenoids have biological functions

protecting bio-membranes from degradation and are involved in the formation of functional bacterial membrane microdomains [14]. Moreover, carotenoids are the constituents of antenna pigments in photosynthesis and protect against oxidative stress [15]. As provitamin A, carotenoids alleviate the disease caused by vitamin A deficiency [16].

Intriguingly, carotenoids participate in response mechanisms to different stressors, like dark-chilling modification of galactolipid and carotenoid composition during chloroplast biogenesis in cucumber cotyledons [17]. Lutein and zeaxanthin contents increase with the intensification of NaCl stress in yellow corn and further increase in supplemental $CaCl_2$ [18]. Under heat stress conditions, the heat-tolerant genotypes BG 240 and JG 14 maintain low levels of violaxanthin in chickpea [19]. Additionally, lutein epoxide is a minor component of the total lutein pool associated with thermal energy dissipation and nonphotochemical quenching during warm months [20]. Foremost, the contents of carotenoids changes in pepper leaves under different temperatures and light intensities [21–23]. Due to the presence of carotenoids in pepper leaves, these properties are closely correlated with antioxidant capacity and tolerance mechanisms, which affect the metabolic system of hormones and pigments and further influence the conduction of pepper fruit. However, more studies need to determine the role carotenoids play in pepper leaves when the plant is under abiotic stress. Moreover, studies should identify the optimal parameters in which to extract carotenoids effectively.

Concerning ultrasound-assisted extraction (UAE), carotenoids can be extracted in less time, at lower temperatures, with less energy and solvent requirements [24], as a non-thermal extraction technique is better equipped to retain the functionality of the bioactive compounds, and is suitable for extracting carotenoids. Concerning solvent acetone and ethyl acetate (1:2), a strong polarity mixed solvent chosen in the study, there is a certain effect on the efficient extraction of carotenoids which belong to strong polarity molecules (Figure 1), but supercritical fluid extraction (SFE) is inefficient for the extraction of highly polar carotenoids and is not suitable for samples with high moisture contents, moreover, the cost of equipment is high. However, the variables associated with UAE such as temperature, time, solvent type, and liquid–solid ratio needs to be optimized for each by-product.

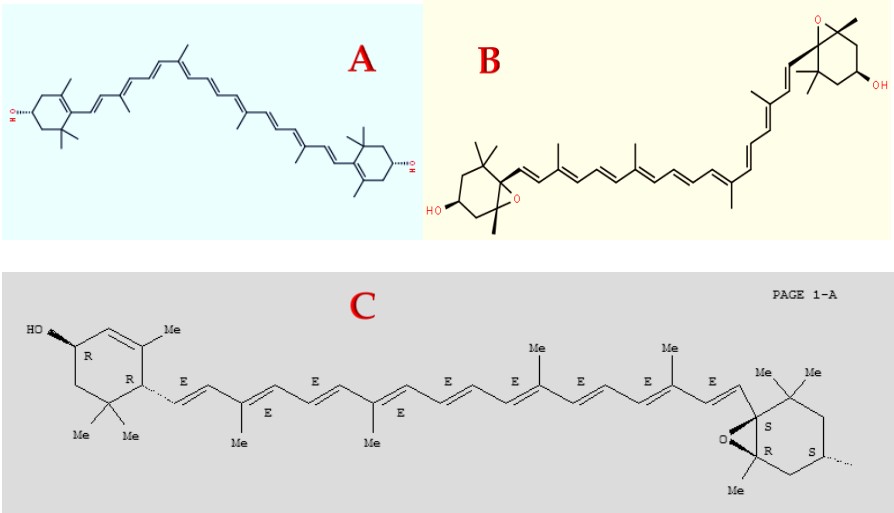

**Figure 1.** Molecular structure of zeaxanthin, violaxanthin, and lutein epoxide. Note: (**A**–**C**) represent zeaxanthin, violaxanthin, and lutein epoxide.

The experimental design was based on the signal factor experiment [5] in which the solution ultrasonic time, solid–liquid ratio, saponification time, and KOH–methanol solution concentrations significantly affected the contents of zeaxanthin, lutein epoxide, and violaxanthin, and the experiment was aimed to verify the optimal factors for carotenoid extraction in multi-response optimization. Moreover, response surface methodology (RSM), a method combining statistics and mathematics, is extensively employed to identify and

solve relationships between random variables and system responses of complex systems. In RSM, comprehensive statistical experimental techniques are used to assess the effects of multiple factors and their interactions on one or more response variables [25–27]. Hence, RSM combined with UAE is widely applied in extracting antioxidant compounds relevant to food science, technology, and medicine [28]. For example, RSM was adopted to optimize the required proportions of sodium benzoate, potassium sorbate, and anthocyanins from purple sweet potato against *Geotrichum candidum* [29] by UAE. In another study [30], RSM was employed to extract anthocyanins from grape juice waste by using microwave-assisted extraction (MAE) at various microwave powers, exposure times, and solvent–solid ratios to identify bioactivity of anthocyanins in cell systems. Liaudanskas [31] utilized RSM to optimize temperature, extraction time, and ultrasonic power to obtain the highest extraction yield of flavonoids from lyophilized apple samples. In peanuts [32], RSM was used to optimize ultrasound-assisted extraction conditions, like sample-to-extractant ratios, sonication times, and sonication temperatures for aflatoxin B1 to ensure the content of antioxidant [33,34]. Therefore, we employed the RSM to optimize the extraction conditions for zeaxanthin, lutein epoxide, and violaxanthin, which were based on our preliminary experiments. Furthermore, with regard to optimizing extraction factors with RSM, total carotenoids extracted from Aresch [35] by subcritical fluid extraction was 0.239 g/kg, zeaxanthin and lutein from corn gluten by UAE were separated and purified using silica gel column chromatography with the purity of zeaxanthin increasing from 0.28% to 31.5% (about 110 times) and lutein from 0.25% to 16.3% (about 65 times) [36]. Using the high hydrostatic-pressure-assisted extraction method, $2.01 \pm 0.09$ mg/100 g of lycopene was obtained from the tomato [37], so, different methods with various species contributed several yields. However, studies attempting to optimize the extraction conditions by UAE with RSM for carotenoids in pepper leaves, thus far, have been less reported. Therefore, factors relevant to extraction and purification, such as sonication time, sample-to-solvent ratio, saponification time, and concentration of saponification solution, must be optimized to obtain higher carotenoid yields.

In this study, acetone and ethyl acetate (1:2) was adopted to extract three carotenoid compositions including zeaxanthin, lutein epoxide, and violaxanthin from pepper leaves, which was selected from our previous work [38]. Equal amounts of solvents, including isopropanol, ethanol, ethyl acetate, acetone, and petroleum were separately added to extract carotenoid compositions. The results showed that acetone and ethyl acetate were the best two solvents for extraction, and further study of the different ratios (3:1, 2:1, 1:1, 1:2, and 1:3) of the two were performed, where the results indicated that acetone:ethyl acetate = 2:1 was the best choice. In addition, the contents of zeaxanthin, lutein epoxide, and violaxanthin associated with the tolerance mechanisms to low-temperature combined with low-light conditions has not been studied extensively. Moreover, this study was concerned about optimum parameters for extracting three kinds of carotenoids from pepper leaves by response surface methodology, aiming to determine the optimum parameters for extracting three carotenoids from pepper leaves, which provide a theoretical basis and method for cultivating new varieties resistant to low temperatures and weak light.

## 2. Materials and Methods

### 2.1. Plants and Growth Conditions

The experiment was conducted at Gansu Agricultural University in Lanzhou (N 36°05′39.86″, E 103°42′31.09″). Pepper seeds of 'Long jiao No. 5' (*Capsicum annuum* L., from Chinese Academy of Agricultural Sciences) were germinated and planted in black plastic pots (9 cm × 9 cm) filled with a seedling-raising substrate (vermiculite:grass carbon:cow dung = 3:1:1). Following regular cultivation management, two seeds were sown and grown with average day/night temperatures of 25 °C/15 °C under natural light (approximately 300 μ mol m$^{-2}$ s$^{-1}$) at a relative humidity of 60–70% in a plant growth chamber (Ningbo Southeast Instrument, Ningbo, China).

The seedlings were grown until they had seven fully expanded leaves (about 50 days after sowing). The third and fourth leaves were randomly selected, cleaned with distilled water of which 2 g each was weighed and wrapped quickly in liquid nitrogen, and stored in a refrigerator at −80 °C for later analysis.

## 2.2. Reagents and Chemicals

Zeaxanthin, lutein epoxide, and violaxanthin were purchased from Sigma (Burbank, CA, USA). Acetone, ethyl acetate, methanol, KOH, butylated hydroxytoluene (BHT), quartz sand, 50 mL centrifuge tube and 0.22 μm organic filter film were purchased from Sinopharm Group Chemical Reagent (Shanghai, China), 2 mL brown chromatographic flasks were obtained from the USA Agilent, all other chemicals used were chromatographic grade.

## 2.3. HPLC Analytical Conditions

Carotenoids were separated using a Waters Alliance high-performance liquid chromatographic equipment consisting of a Waters 2695 Separation Module (Waters, Milford, MA, USA). The HPLC system had a column oven, an automatic sampler, and an online degasser and was equipped with a Waters 2487 dual λ absorbance ultraviolet detector. The detection wavelength was 450 nm, and the best temperature for the column was 30 °C. About 20 μL volume was injected into the system. The mobile phase consisted of four groups: (A) acetonitrile; (B) water; (C) methyl tert-butyl ether: methanol (1:1, *v:v*); and (D) ethyl acetate. Operation and data analysis was conducted using Empower software.

## 2.4. Experimental Design for Carotenoid Extraction

The response surface experiment was designed according to the principle of RSM on the basic of a preliminary single-factor experiment [5] (the ultrasound time was 40 min, ratio of solid–liquid was 1:8, saponification for 30 min, and 20% KOH–methanol solution), the upper and lower limits were set based on this standard, a total of five graded as independent variables, with carotenoids content set as response value. A four-factor inscribed central composite design (CCD) was used to identify the relationship between the response and the variables, as well as to determine variables that optimized the extraction process of the three carotenoids contents. Each variable was coded with the minimum, the best and the maximum level (−1, 0, 1) given in Table 1:

**Table 1.** Experimental design level.

| Level | Ultrasound Time min (A) | Solid–Liquid Ratiog/mL (B) | Saponification Time min (C) | Saponification Solution Concentration V % (D) |
|---|---|---|---|---|
| −1 | 20 | 1/12 | 10 | 10 |
| 0 | 40 | 1/8 | 30 | 20 |
| 1 | 60 | 1/4 | 50 | 30 |

Frozen pepper leaves (2.0 g) were ground with 0.1 g BHT (to prevent the samples from oxidation) and liquid nitrogen until they became a homogenate mixture. KOH–methanol solutions of appropriate concentrations (10%, 15%, 20%, 25%, and 30%, *v/v*) were added to the mixture to a final volume of 8 mL. Then, the mixture was quickly transferred to a 50 mL centrifuge tube. The temperature of the mixture was kept constant at 55 °C in a thermostatic water bath (Shanghai Yuejin medical instrument, Shanghai, China) for 10, 20, 30, 40, and 50 min, after which the mixture was immediately chilled with cold water. According to the suitable solid-to-liquid ratios (1:4, 1:6, 1:8, 1:10, and 1:12), 16 mL of an acetone–ethyl acetate (1:2, *v/v*) extracting solution was added to a 50 mL centrifuge tube. Afterward, the mixture was extracted for 20, 30, 40, 50, and 60 min on an ultrasonic cleaner (Ningbo Scientz Biotechnology, Ningbo, China) and centrifuged for about 15 min at 4 °C at 8000 r/min in a high-speed refrigerated centrifuge (USA Sigma). The concentrated supernatant was dried using a rotary evaporator (Shanghai Yarong biochemical instrument

factory, Shanghai, China) for about 5 min. The acetone used to determine the volume was 10 mL. The 1.5 mL upper liquid was filtered into a chromatography flask through a 0.22 μm organic filter membrane storage in a refrigerator at −80 °C for the next measurements of carotenoids by HPLC. Given that carotenoids are sensitive to temperature and light, in all extractions progress, samples were placed in ice and operated after 6 pm to avoid bright light.

Twenty-nine random experiments (four repeat for every treatment) were assigned based on CCD and the values of independent variables were considered. Depicted in Table 2, A is ultrasound time (min), B is solid–liquid ratio (g/mL), C is saponification time (min), and D is saponification solution concentration (V%).

**Table 2.** Table and results of response surface analysis.

| Run Order | A | B | C | D |
|---|---|---|---|---|
| 1 | 30 | 1:6 | 40 | 25 |
| 2 | 40 | 1:8 | 30 | 20 |
| 3 | 60 | 1:8 | 30 | 20 |
| 4 | 50 | 1:10 | 40 | 15 |
| 5 | 30 | 1:10 | 20 | 25 |
| 6 | 40 | 1:8 | 30 | 10 |
| 7 | 50 | 1:6 | 20 | 15 |
| 8 | 30 | 1:6 | 40 | 15 |
| 9 | 50 | 1:6 | 40 | 15 |
| 10 | 30 | 1:6 | 20 | 15 |
| 11 | 30 | 1:10 | 40 | 15 |
| 12 | 40 | 1:4 | 30 | 20 |
| 13 | 40 | 1:8 | 10 | 20 |
| 14 | 40 | 1:12 | 30 | 20 |
| 15 | 30 | 1:10 | 40 | 25 |
| 16 | 40 | 1:8 | 50 | 20 |
| 17 | 50 | 1:10 | 20 | 25 |
| 18 | 50 | 1:10 | 40 | 25 |
| 19 | 40 | 1:8 | 30 | 20 |
| 20 | 50 | 1:10 | 20 | 15 |
| 21 | 20 | 1:8 | 30 | 20 |
| 22 | 40 | 1:8 | 30 | 20 |
| 23 | 30 | 1:6 | 20 | 25 |
| 24 | 40 | 1:8 | 30 | 20 |
| 25 | 40 | 1:8 | 30 | 30 |
| 26 | 50 | 1:6 | 20 | 25 |
| 27 | 40 | 1:8 | 30 | 20 |
| 28 | 30 | 1:10 | 20 | 15 |
| 29 | 50 | 1:6 | 40 | 25 |

## 2.5. Validation of Working Curves and Standard Solutions

To prepare the standard solutions, 5 mg of zeaxanthin, lutein epoxide, and violaxanthin were weighed and dissolved in acetone and diluted to 100 mg/L and stored at −18 °C for later experiments. Linear regression equations were established by taking the concentration of zeaxanthin, lutein epoxide, and violaxanthin as the *X*-axis and the peak area as the Y-axis, with the standard curve referencing Li et al. [5].

## 2.6. Determination of Carotenoid Content

The contents of three carotenoids zeaxanthin, lutein epoxide, and violaxanthin, were calculated by determining the working curves of the standard solutions.

*2.7. Statistical Analysis*

Data were analyzed using Microsoft Excel, other experiment design, second order polynomial model, table of analysis of variance, and contour and surface diagrams were from Design-Expert 8.0.6.

**3. Results**

*3.1. Carotenoid Contents*

According to the optimal conditions of A, 40 min, B, 1:8 g/mL, C, 30 min, D, 20%, actual extracted values of zeaxanthin, lutein epoxide, and violaxanthin had a maximum extraction content of 0.8118, 3.9497, and 16.1590 µg/g, respectively. Notably, five groups had the same maximum extraction content (Table 3).

**Table 3.** Table and results of carotenoid content.

| Order | Zeaxanthin µg/g | Lutein Epoxide µg/g | Violaxanthin µg/g |
|---|---|---|---|
| 1 | 0.6458 | 0.7242 | 9.2004 |
| 2 | 0.8118 | 3.9497 | 16.1590 |
| 3 | 0.6297 | 0.3069 | 4.1704 |
| 4 | 0.6497 | 1.6688 | 9.5840 |
| 5 | 0.7290 | 0.3723 | 2.6644 |
| 6 | 0.4750 | 0.7214 | 9.7940 |
| 7 | 0.1640 | 0.3456 | 8.6202 |
| 8 | 0.5035 | 1.0197 | 6.1351 |
| 9 | 0.4063 | 0.3554 | 6.3390 |
| 10 | 0.8059 | 1.0344 | 9.2515 |
| 11 | 0.6928 | 3.1546 | 11.3549 |
| 12 | 0.2260 | 0.4184 | 3.7583 |
| 13 | 0.5273 | 1.2497 | 9.1722 |
| 14 | 0.7886 | 0.5488 | 8.5340 |
| 15 | 0.6631 | 0.9658 | 8.1411 |
| 16 | 0.4951 | 2.2707 | 5.7207 |
| 17 | 0.5398 | 1.8681 | 13.5851 |
| 18 | 0.4673 | 3.8501 | 11.4879 |
| 19 | 0.8118 | 3.9497 | 16.1590 |
| 20 | 0.2683 | 1.2786 | 12.5240 |
| 21 | 0.5922 | 0.7547 | 7.7989 |
| 22 | 0.8118 | 3.9497 | 16.1590 |
| 23 | 0.5393 | 0.7326 | 9.6992 |
| 24 | 0.8118 | 3.9497 | 16.1590 |
| 25 | 0.5270 | 0.4200 | 11.7098 |
| 26 | 0.2153 | 0.3043 | 1.9771 |
| 27 | 0.8118 | 3.9497 | 16.1590 |
| 28 | 0.3669 | 0.5481 | 10.4746 |
| 29 | 0.1835 | 0.2056 | 15.0405 |

*3.2. Variance and Significance Analysis of Regression Model*

A second-order polynomial model was adopted for the fitting analysis of the data to obtain a function of the zeaxanthin, lutein epoxide and violaxanthin extraction yield using Design Expert 8.0.6:

$$R_1 = 0.81 - 0.082A + 0.085B + 0.022C + 9.575D + 0.062AB + 0.028AC - 0.018AD + 0.035BC + 0.045BD - 0.044CD - 0.057A^2 - 0.083B^2 - 0.082C^2 - 0.085D^2$$

$$R_2 = 3.95 + 0.018A + 0.39B + 0.31C - 0.041D + 0.37AB - 0.056AC + 0.35AD + 0.36BC + 0.075BD - 0.033CD - 0.80A^2 - 0.81B^2 - 0.49C^2 - 0.79D^2$$

$$R_3 = 16.16 + 0.21A + 0.96B + 0.066C + 0.056D + 1.05AB + 0.19AC + 0.78AD - 0.37BC - 0.85BD + 1.46CD - 2.29A^2 - 2.25B^2 - 1.92C^2 - 1.09D^2.$$

where the absolute values of $R_1$, $R_2$, $R_3$ are the predicted extraction values of zeaxanthin, lutein epoxide and violaxanthin (µg/g), A is ultrasound time (min), B is the solid–liquid ratio (g/mL), C is saponification time (min), and D is saponification solution concentration (V %). When calculating the predicted content of the three carotenoids ($R_1$, $R_2$, $R_3$), brought in the corresponding variables value (A, B, C, D) separately.

An analysis of variance (ANOVA) was employed to determine the regression coefficients, statistical significance of the model terms, and to fit the mathematical models of the experimental data that aimed to optimize the overall region for both response variables. The results of variance analysis and simulation reliability analysis of the regression equations are shown in Tables 4–6. The *p*-value (assumptions value, less than 0.05 is significant in ANOVA) of lutein epoxide, zeaxanthin and violaxanthin was less than 0.05 in regression model, indicating the model has significant differences, and could be used to predict the response value. Otherwise, the lack of fit item of a *p*-value and an associated F-value > 0.05 implies that lack of fit is not significant relative to the pure error, suggesting that the model had sufficient resolution could reflect the experimental results better. Therefore, this model can be used to analyze and predict the extraction process conditions of zeaxanthin, lutein epoxide and violaxanthin.

The factors in Table 4 affecting the extraction of zeaxanthin from the leaves of pepper were ranked in the following order according to their F-value: B (liquid–solid ratio) > A (ultrasonic time) > C (saponification time) > D (saponification liquid concentration). A, B, $B^2$, $D^2$ and $C^2$ were the significant influencing factors, indicating factors that would serve model better. In addition, the F-value of the lack of fit in the table was 2.960 (more than 0.05), and $R^2 = 0.7595$.

In Table 5, the factors affecting the extraction of lutein epoxide was ranked in the following order according to the F-value: B > C > D > A. $A^2$, $B^2$, and $D^2$ were the extremely significant influencing factors, whereas B, AB, AD, BC, and $B^2$ were the significant influencing factors indicating those factors that serve the model better. In addition, the F-value of the lack of fit in the table was 3.5214 (more than 0.05) and $R^2 = 0.9024$.

In Table 6, the factors affecting the extraction of violaxanthin were ranked in the following order according to their F-value: D > B > A > C. $A^2$, $B^2$, and $C^2$ were the significant factors, indicating those factors that serve the model better. In addition, the F-value of the lack of fit in the table was 1.29457 (more than 0.05), and $R^2 = 0.7384$.

**Table 4.** Variance and significance analysis of regression model for zeaxanthin.

| Source | Sum of Squares | Df | Mean Square | f-Value | *p*-Value Prob > f | Difference Significant |
|---|---|---|---|---|---|---|
| Model | 0.90 | 14 | 0.064 | 3.16 | 0.0197 | ** |
| A | 0.15 | 1 | 0.15 | 7.36 | 0.0169 | ** |
| B | 0.16 | 1 | 0.16 | 7.84 | 0.0142 | ** |
| C | 0.015 | 1 | 0.015 | 0.73 | 0.4069 | |
| D | $9.428 \times 10^{-4}$ | 1 | $9.428 \times 10^{-4}$ | 0.046 | 0.8329 | |
| AB | 0.073 | 1 | 0.073 | 3.56 | 0.0801 | |
| AC | $8.837 \times 10^{-3}$ | 1 | $8.837 \times 10^{-3}$ | 0.43 | 0.5211 | |
| AD | $2.784 \times 10^{-3}$ | 1 | $2.784 \times 10^{-3}$ | 0.14 | 0.7174 | |
| BC | 0.025 | 1 | 0.025 | 1.23 | 0.2861 | |
| BD | 0.025 | 1 | 0.025 | 1.25 | 0.2829 | |
| CD | 0.025 | 1 | 0.025 | 1.22 | 0.2876 | |
| $A^2$ | 0.088 | 1 | 0.088 | 4.30 | 0.0571 | |
| $B^2$ | 0.18 | 1 | 0.18 | 8.98 | 0.0096 | ** |
| $C^2$ | 0.18 | 1 | 0.18 | 8.78 | 0.0103 | ** |
| $D^2$ | 0.19 | 1 | 0.19 | 9.32 | 0.0086 | ** |
| Residual | 0.29 | 14 | 0.020 | | | |
| Lack of Fit | 0.29 | 10 | 0.029 | $2.960 \times 10^{+8}$ | <0.0001 | *** |
| Pure Error | $3.859 \times 10^{-10}$ | 4 | $9.649 \times 10^{-11}$ | | | |
| Cor Total | 1.19 | 28 | | | | |
| $R^2$ | 0.7595 | | | | | |

Note: Significant difference is shown with an asterisk in the next three tables; extremely significant is "***", significant is "**".

**Table 5.** Variance and significance analysis of regression model for lutein epoxide.

| Source | Sum of Squares | Df | Mean Square | F-Value | *p*-Value Prob > F | Difference Significant |
|---|---|---|---|---|---|---|
| Model | 49.40 | 14 | 3.53 | 9.24 | <0.0001 | *** |
| A | 7.679 | 1 | 7.679 | 0.020 | 0.8892 | |
| B | 3.56 | 1 | 3.56 | 9.33 | 0.0086 | ** |
| C | 2.35 | 1 | 2.35 | 6.14 | 0.0265 | |
| D | 0.040 | 1 | 0.040 | 0.11 | 0.7497 | |
| AB | 2.19 | 1 | 2.19 | 5.75 | 0.0310 | ** |
| AC | 0.050 | 1 | 0.050 | 0.13 | 0.7231 | |
| AD | 1.92 | 1 | 1.92 | 5.03 | 0.0417 | ** |
| BC | 2.02 | 1 | 2.02 | 5.29 | 0.0374 | ** |
| BD | 0.089 | 1 | 0.089 | 0.23 | 0.6363 | |
| CD | 0.017 | 1 | 0.017 | 0.045 | 0.8353 | |
| $A^2$ | 16.70 | 1 | 16.70 | 43.73 | <0.0001 | *** |
| $B^2$ | 17.19 | 1 | 17.19 | 45.02 | <0.0001 | *** |
| $C^2$ | 6.35 | 1 | 6.35 | 16.64 | 0.0011 | ** |
| $D^2$ | 16.28 | 1 | 16.28 | 42.65 | <0.0001 | *** |
| Residual | 5.35 | 14 | 0.38 | | | |
| Lack of Fit | 5.35 | 10 | 0.53 | 3.5214 | 0.052 | |
| Pure Error | 1.26 | 4 | 0.24 | | | |
| Cor Total | 54.75 | 28 | | | | |
| $R^2$ | 0.9024 | | | | | |

**Table 6.** Variance and significance analysis of regression model for violaxanthin.

| Source | Sum of Squares | Df | Mean Square | F-Value | *p*-Value Prob > F | Difference Significant |
|---|---|---|---|---|---|---|
| Model | 364.48 | 14 | 26.03 | 2.82 | 0.0309 | ** |
| A | 1.03 | 1 | 1.03 | 0.11 | 0.7428 | |
| B | 22.24 | 1 | 22.24 | 2.41 | 0.1427 | |
| C | 0.10 | 1 | 0.10 | 0.011 | 0.9167 | |
| D | 0.075 | 1 | 0.075 | 8.163 | 0.9293 | |
| AB | 17.76 | 1 | 17.76 | 1.93 | 0.1870 | |
| AC | 0.56 | 1 | 0.56 | 0.061 | 0.8083 | |
| AD | 9.82 | 1 | 9.82 | 1.06 | 0.3196 | |
| BC | 2.14 | 1 | 2.14 | 0.23 | 0.6377 | |
| BD | 11.61 | 1 | 11.61 | 1.26 | 0.2807 | |
| CD | 34.23 | 1 | 34.23 | 3.71 | 0.0746 | |
| $A^2$ | 135.54 | 1 | 135.54 | 14.70 | 0.0018 | ** |
| $B^2$ | 130.80 | 1 | 130.80 | 14.18 | 0.0021 | ** |
| $C^2$ | 95.67 | 1 | 95.67 | 10.37 | 0.0062 | ** |
| $D^2$ | 31.04 | 1 | 31.04 | 3.37 | 0.0879 | |
| Residual | 129.12 | 14 | 9.22 | | | |
| Lack of Fit | 129.12 | 10 | 12.91 | 1.29457 | 0.0687 | |
| Pure Error | 1.08 | 4 | 0.47 | | | |
| Cor Total | 493.6 | 28 | | | | |
| $R^2$ | 0.7384 | | | | | |

In summary, the data demonstrated that the model had a high fitting degree and few experimental errors. Hence, the model could be used to analyze and predict the effects of optimizing the extraction of zeaxanthin, antheraxanthin, and violaxanthin from pepper leaves.

### 3.3. Analysis of Response Surface Optimization Test on Contour and Surface Diagrams

After analysis of variance (ANOVA), response surface analysis was used to determine the effect of independent variables on the average extraction yield of three carotenoids again. The ordinate and the abscissa represent the extraction yields and any two variables, respectively. The three-dimensional profiles indicated how any two variables influenced the yield, as well as the two-dimensional contour lines graph demonstrate clearly the optimal yield under any two variables. The effects of ultrasound time (A), solid–liquid ratio (B), saponification time (C), and saponification solution concentration (D) on the extraction yield are shown in Figures 2–4.

The extraction yield of zeaxanthin gradually increased with A, B, C, and D at approximately 30 to 40 min, 1:10 to 1:8, 20 to 32 min and 15% to 20%, respectively. Further increasing parameters led to a decrease in the extraction of zeaxanthin. The surfaces have obvious upper convex in Figure 2d,f,h,j,l, and a slight upper convex in Figure 1b with a maximum value at the center of the response surface, which confirm the rationality of the experimental models. In six contour lines, the response value points in (Figure 2c,e,g,i,k) were inside the first contour, but Figure 1a was approximately on the first contour. Based on the multivariate regression fitting equation, the optimized extraction conditions were obtained: A, 39.56 min; B, 8.17:1; C, 30.71 min; D,18.7%, resulting in a predicted extraction yield of 0.8230 μg/g.

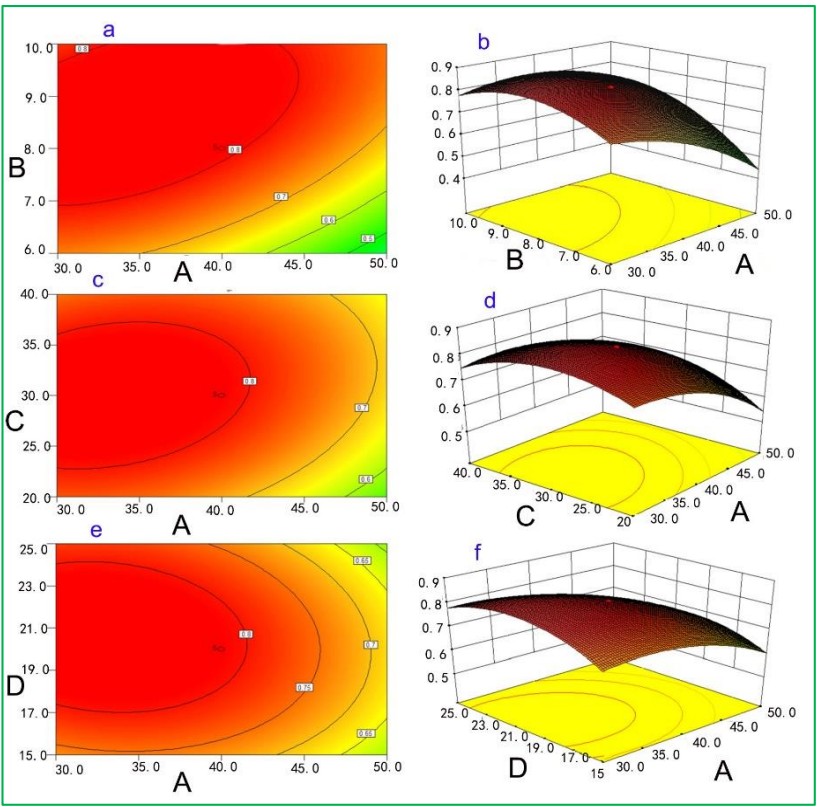

**Figure 2.** *Cont*.

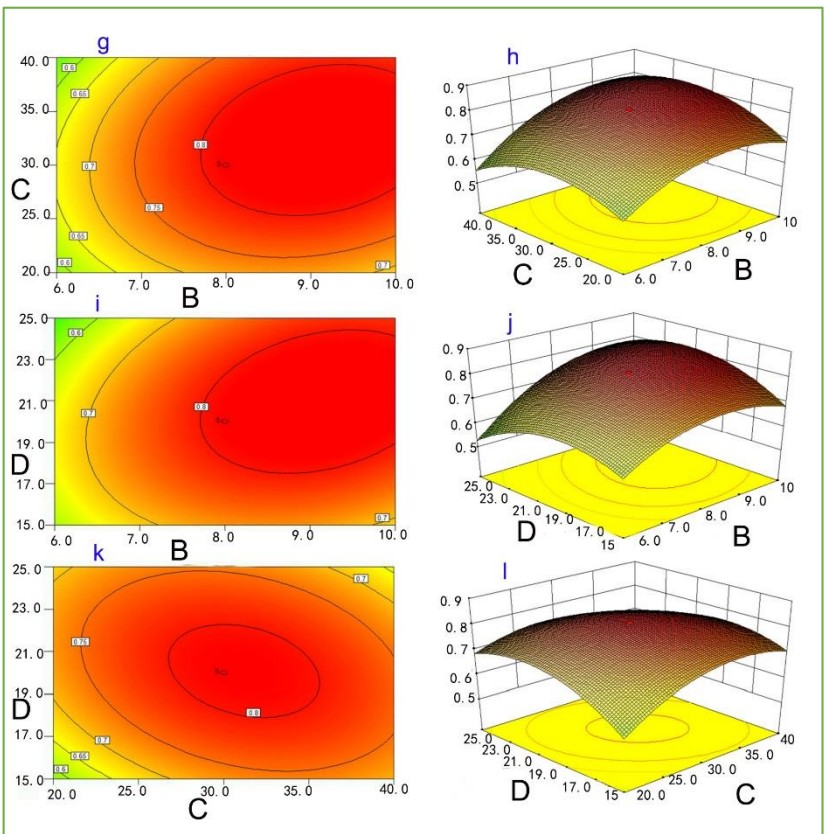

**Figure 2.** Contour diagrams and response surface curves for the effects of (**a**,**b**) ultrasound time and solid–liquid ratio, (**c**,**d**) ultrasound time and saponification time, (**e**,**f**) ultrasound time and saponification solution concentration, (**g**,**h**) solid–liquid ratio and saponification time, (**i**,**j**) solid–liquid ratio and saponification solution concentration, (**k**,**l**) saponification time and saponification solution concentration on the extraction yield of zeaxanthin.

In Figure 3, the extraction yield of lutein epoxide gradually increased with A, B, C, and D at approximately 30 to 40 min, 1:10 to 1:8, 20 to 32 min and 15% to 20%, respectively. Further increased parameters led to a decrease in the extraction of zeaxanthin. The surfaces have obvious upper convex in Figure 3b,f,h,j,l, and a slight upper convex in Figure 2d with a maximum value at the center of the response surface, which confirms the rationality of the experimental models. In six contour lines, the response values (Figure 2a,e,g,i,k) corresponding to the first contours were oval, but Figure 3c was approximately circular. Based on the multivariate regression fitting equation, the optimized extraction conditions were obtained: A, 39.53 min; B, 8.03:1; C, 30.46 min; D, 20.71%, resulting in a predicted extraction yield of 4.0368 μg/g.

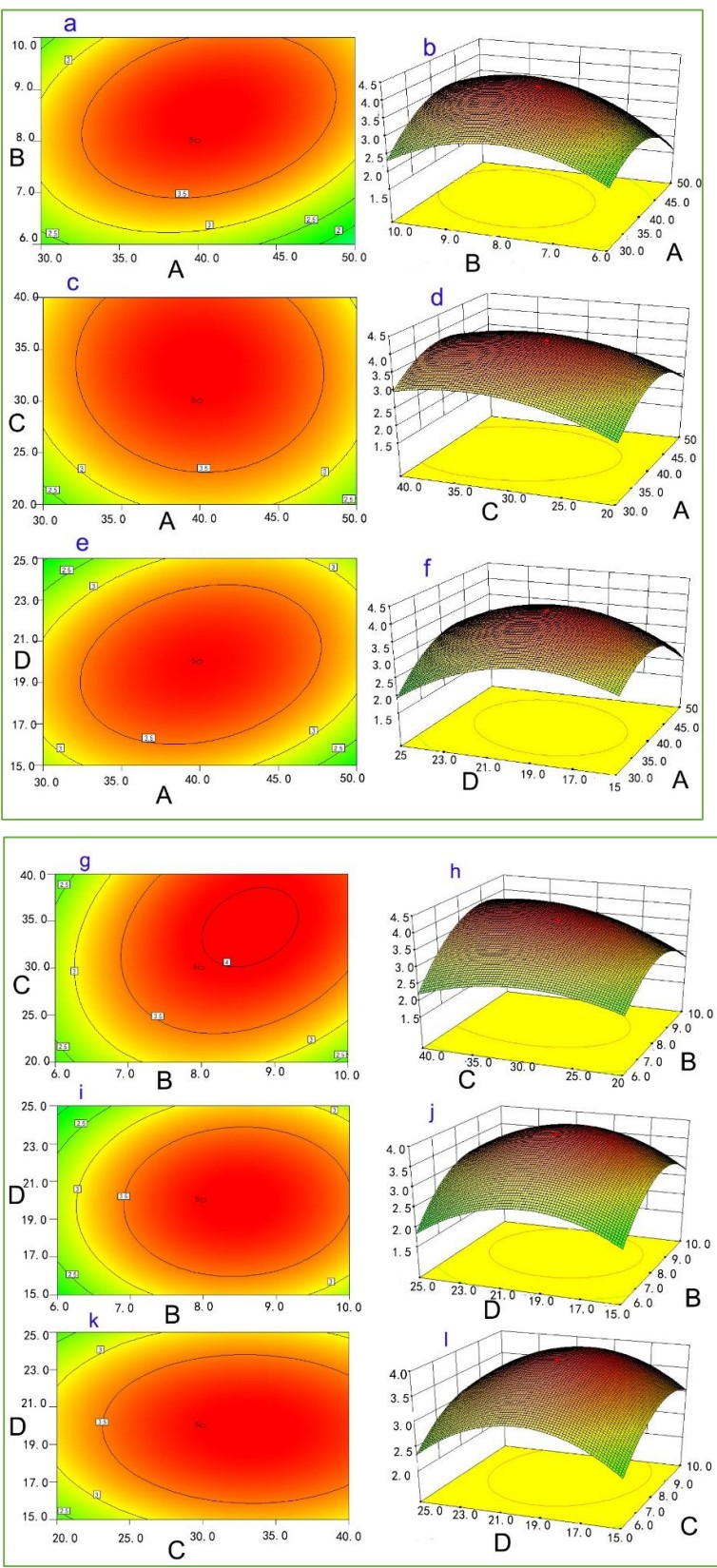

**Figure 3.** Contour diagrams and response surface curves for the effects of (**a**,**b**) ultrasound time and solid–liquid ratio, (**c**,**d**) ultrasound time and saponification time, (**e**,**f**) ultrasound time and saponification solution concentration, (**g**,**h**) solid–liquid ratio and saponification time, (**i**,**j**) solid–liquid ratio and saponification solution concentration, (**k**,**l**) saponification time and saponification solution concentration on the extraction yield of lutein epoxide.

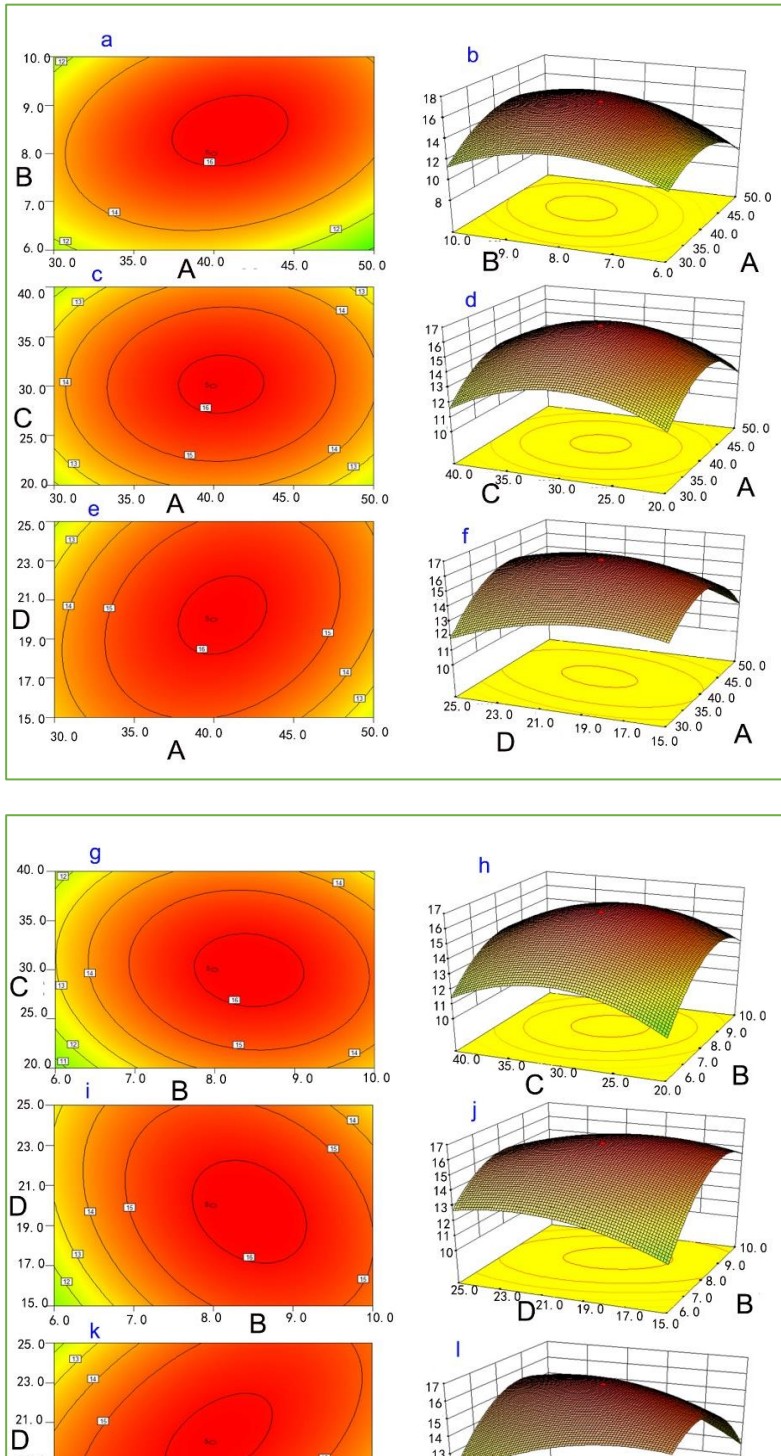

**Figure 4.** Contour diagrams and response surface curves for the effects of (**a**,**b**) ultrasound time and solid–liquid ratio, (**c**,**d**) ultrasound time and saponification time, (**e**,**f**) ultrasound time and saponification solution concentration, (**g**,**h**) solid–liquid ratio and saponification time, (**i**,**j**) solid–liquid ratio and saponification solution concentration, (**k**,**l**) saponification time and saponification solution concentration on the extraction yield of violaxanthin.

In Figure 4, the extraction yield of violaxanthin gradually increased with A, B, C, and D at approximately 30 to 40 min, 1:10 to 1:8, 20 to 32 min and 15% to 20%, respectively. Further, parameters increase led to a decrease in the extraction of zeaxanthin. The surfaces have obvious upper convexes in Figure 4b,d,f,h,l, and a slight upper convex in Figure 3j with a maximum value at the center of the response surface, which confirmed the rationality of the experimental models. In six contour lines, the response value points (Figure 4a,c,e,g,i,k) were all inside the first contour, and the first contours were oval. Based on the multivariate regression fitting equation, the optimized extraction conditions were obtained: A, 40.26 min; B, 8.07:1; C, 30.01 min; D, 20.05%; resulting in a predicted extraction yield of 16.1972 µg/g.

### 3.4. Optimum Extraction Process

According to Design-Expert 8.0.6, the optimal conditions for ultrasonic extraction of zeaxanthin, lutein epoxide, and violaxanthin from pepper leaves were 40.38 min of ultrasonic time, a solid-to-liquid ratio of 1:8.34, saponification time of 32.16 min, and saponification liquid volume fraction of 20.60% KOH–methanol solution. The predicted amount of zeaxanthin, lutein epoxide, and violaxanthin extraction were 0.823022, 4.03684, and 16.1972 µg/g, respectively. To validate the results of the precision of the model, the four conditions were revised to 40 min, 1:8, 30 min and 20% given the feasibility of actual operation.

### 3.5. Validation Test

For the sake of validating the fitting degree of the model by RSM, the optimum paraments experiments were repeated four times. As shown in Table 7, the predicted value originated from the "R" of the second order polynomial models, the actual value from the validated experiment, and the three carotenoids' relative deviations were less than 2.2%, suggesting that the simulation of the carotenoid response surface fitting model worked well.

Relative deviation (%) = (Predict value − Actual value)/Predict value * 100%

**Table 7.** Verified test results of the carotenoid response surface fitting model.

| Average Data | Extraction of Carotenoids (µg/g) | | |
|---|---|---|---|
| | **Zeaxanthin** | **Lutein Epoxide** | **Violaxanthin** |
| Predict value | 0.8230 | 4.0368 | 16.1972 |
| Actual value | 0.8118 | 3.9497 | 16.1590 |
| Relative deviation(%) | 1.36 | 2.16 | 0.24 |

## 4. Discussion and Conclusions

### 4.1. Discussion

Zeaxanthin and violaxanthin are key precursors of light-harvesting carotenoids involved in the photoprotective xanthophyll cycle, while lutein epoxide is unique to photosynthetic eukaryotes [39,40]; they are absolutely necessary in biological activities. At the same time, the extracted carotenoids are significant in the research field too. As the first vital step, extraction parameters stand in an important position in extracting carotenoids from the leaves of pepper seedlings by UAE. Three kinds of carotenoids were extracted from pepper leaves with acetone–ethyl acetate (1/1, *v/v*) in this experiment. The solid–liquid ratio was 1:8, and the ultrasonic time was 40 min at 30 °C. There were inconsistent and consistent conclusions with relevant studies.

The extraction parameters of the solid–liquid ratio and ultrasonic time used more time and dissipated more energy, thereby oxidizing and denaturing small molecule extracts. A study showed that zeaxanthin can be ultrasonically extracted with high efficiency from corn gluten meal by ethanol at a liquid–solid ratio of 7.9:1 at an extraction time of 45 min at 56 °C [36] via UAE, with a higher temperature. Similarly, Nasir showed that ultrasonic

extraction of antioxidant compounds from *Chlorella vulgaris* uses more time and extraction solution, the optimal conditions were an extraction time of 146 min, ethanol as the solvent for extraction, and a liquid ratio of 62 mL/g [41]. Carotenoids extracted from red pepper fruit by SFE after 21 days in cool storage (7 °C) [3], the carotenoid concentrations expressed in μg/g of the edible portion were: zeaxanthin (8.53), more than that extracted by UAE from pepper leaves; violaxanthin (7.70), less than carotenoids from pepper leaves, which is because the carotenoid content of fruits and vegetables is affected by many factors such as the variety, ripeness, climate, geographic site of production, the part of the plant used, environmental conditions during agricultural production, postharvest handling, processing and storage conditions. Similarly, other studies on plant foods such as strawberry and peppers also described an increase in the concentration of antioxidants after a period of time in cold storage. Therefore, the different content of carotenoids may be caused by cold storage, but the model could explain that RSM is suitable for effective extraction of carotenoids from pepper leaves. Another study investigated the optimized extraction conditions for total phenolics and carotenoids from the leaves of *Centella asiatica*, the optimum extracted solution of ethanol concentration, extraction time for carotenoids were 100% for 110.5 min [42]. These diverging results were obtained because of the different properties of the extraction materials used and different extraction solvents utilized, leading to different conditions for optimal ultrasonic extraction. Moreover, these extraction conditions were not optimal for the extraction of carotenoids, as well as extraction yield.

Saponification also is crucial to carotenoid extraction. It was good at removing the interfering substances during the extraction of the carotenoids, and can remove chlorophyll and esterified fatty acids from pepper leaves used to extract pure carotenoids. A study explored an extract of lutein fatty acid esters from marigold flowers by using supercritical carbon dioxide (SC-$CO_2$) with a co-solvent, and found that saponification of oleoresin with 40% (*w/v*) KOH can convert lutein fatty acid esters into free lutein. Saponification can qualitatively and quantitatively extract pigments, but it can also cause the degradation of carotenoids to a certain extent [43,44]. In the present study, the saponification solution was a KOH–methanol solution with a concentration of 20% and a saponification time of 40 min, a suitable concentration of KOH–methanol with appropriate time causing a better effect. Thus, ultrasonication and saponification played an important role in extracting the carotenoid content from pepper leaves, which was consistent with the objective of this study to optimize the extraction conditions.

Studies have also investigated extracting lycopene, lutein and carotenoids from tomato pulp, paprika leaf and persimmon separately using High Hydrostatic Pressure-assisted Extraction (HHPE), accelerated Solvent Extraction (ASE), Supercritical Fluid Extraction (SFE) [45–47]. However, carotenoid extraction is sensitive to light and temperature, moreover, SFE does not apply oxidation and dissipation of heat-sensitive substances, while the ASE method involves nonpolar solvents like toluene and petroleum ether, and HHPE is limited to highly tolerant matters. Therefore, ultrasonic-assisted solvent extraction is the optimal extraction method for zeaxanthin, lutein epoxide and violaxanthin.

Regarding the response surface analysis and analysis of variance (ANOVA) in this study, the quality of fitness to the second-order polynomial models for leaf extracts of pepper was established based on the coefficients of determination (70% > $R^2$), and the regression *p*-value ($p \leq 0.1$). The "fitness" of the model was studied through the lack-of-fit test ($p \leq 0.05$) indicating the adequacy of models to accurately predict variables and that the models could be used to predict the responses, which is similar to the research conducted by Gunathilake, et.al and Wang, et.al. [36,42]. The determination coefficient of the experimental model $R^2$ of lutein epoxide, zeaxanthin and violaxanthin were above 73%, which means the test fit the model better; 73% of data could be explained by second-order polynomial models, but in comparison to zeaxanthin and violaxanthin, it is sufficient in providing strong support for the optimization of extraction conditions for carotenoids.

*4.2. Conclusions*

In our study, the RSM was successfully implemented to optimize factors of three carotenoids (zeaxanthin, lutein epoxide, and violaxanthin) extracted from pepper leaves. Overall, analysis of the response value results revealed that the optimal extraction conditions of carotenoids from pepper leaves were as follows: an ultrasonic time of 40 min, solid–liquid ratio of 1:8, saponification time of 32 min, and saponification solution (KOH–methanol solution) concentration of 20%. The quality of fitness to the second-order polynomial models and response surface for pepper leaf extracts were established based on the coefficients of determination ($73\% > R^2$), and the regression $p$-value ($p \leq 0.1$), and the "fitness" of the model was studied through the lack of fit test ($p \leq 0.05$). It was revealed that the parameters of the extraction process optimized using the RSM design were reliable and precise. The RSM may provide an experimental basis for subsequent experiments and has great developmental prospects. This research finding will support functional and stress tolerance for the isolation of carotenoids from these leaves, renewing interest in utilizing the leaves of peppers.

**Author Contributions:** N.L. and J.L. contributed to the collection of references and manuscript preparation. Conceptualization, J.L., D.D., J.Z., Y.M., and F.G. Methodology, J.L., N.L., W.L., C.W. and T.N. Writing—review and editing, J.L., E.P.B. and N.L. Funding acquisition, J.L. and J.X. All authors have read and agreed to the published version of the manuscript.

**Funding:** This research was supported by Natural Science Foundation of Gansu Province (20JR10RA515), National Natural Science Foundation of China (32072657), China; the Special Fund for Technical System of Melon and Vegetable Industry of Gansu Province (GARS-GC-1), China; China; the National Key Research and Development Program of China (2016YFD0201005), China.

**Institutional Review Board Statement:** Not applicable.

**Informed Consent Statement:** Not applicable.

**Data Availability Statement:** Data is contained within the article.

**Conflicts of Interest:** The authors declare no conflict of interest.

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
