# Peer review of "Optimum Parameters for Extracting Three Kinds of Carotenoids from Pepper Leaves by Response Surface Methodology"

_separations, doi:10.3390/separations8090134_

Round 1
Reviewer 1 Report
Present research by Nenghui et al. is focused on optimization of ultrasonic-assisted solvent extraction (UAE) of carotenoids from pepper Leaves. Authors used RSM to optimize the extraction process based on maximum content of zeaxanthin, lutein epoxide, and violaxanthin. Carotenoids extraction is very interesting topic and present research should be interesting to Separations readers.
I have already reviewed this paper which was submitted to the Molecules recently and authors have responded to the majority of my comments. Some things MUST still be clarified:
- Add information and explain how did you choose experimental design. Was it literature data, preliminary experiments, etc.? Describe in detail!
- How did you select experimental domain for independent variables?
- Please explain how did you determine model fitness using F-test, coefficient of correlation, lack of fit testing and other statistical parameters.
- Why did you select this method of extraction rather than green technique such as supercritical fluid extraction?
- Please compare your results with literature data on SFE of carotenoids from red pepper.
- Information about desirability function used for multi response optimization is missing. Add that in 4.8. and 2.4. segments. What was the D function value?
- Paper should be proof read by the native English speaker since it has certain grammatical and style errors.
Furthermore, I have an important issue regarding the self-plagiarism. Did you use some results from your previously published work in this manuscript (Li, J., Xie, J., Yu, J., Lv, J., Zhang, J., Wang, X., ... & Ma, G. (2017). Reversed-phase high-performance liquid chromatography for the quantification and optimization for extracting 10 kinds of carotenoids in pepper (Capsicum annuum L.) leaves. Journal of agricultural and food chemistry, 65(38), 8475-8488.)? If yes, please highlight this and cite where necessary.
Reviewer 2 Report
In this paper Authors try to determine the optimum parameters for extracting three carotenoids from pepper leaves by response surface
methodology. Solvent of acetone and ethyl acetate were used to extract carotenoids at four independent factors: sonication time, saponification time, and concentration of saponification solution (KOH–methanol).
It is not clear because Authors have been choosen these extraction methodologies. Authors should justify their choice and explain why other types of solvent, e.g. ethanol, were not considered. In addition, bioactivity tests should be performed to correlate the extraction technique used with the amount of extracted carotenoids and their bioactivity. For example, tests should be performed to evaluate their antioxidant activity in both acellular and cellular systems.
Experimental details have to be added. For example, the temperature of the extraction shoud be specified (lines 138-140).
In my opinion, in this form the manuscrip is not suitable for publication but needs to be improved.
Round 2
Reviewer 2 Report
Authors have improved the manuscript and now it is suitable for publication.
This manuscript is a resubmission of an earlier submission. The following is a list of the peer review reports and author responses from that submission.
Round 1
Reviewer 1 Report
Present research by Nenghui et al. is focused on optimization of ultrasonic-assisted solvent extraction (UAE) of carotenoids from pepper Leaves. Authors used RSM to optimize the extraction process based on maximum content of zeaxanthin, lutein epoxide, and violaxanthin. Carotenoids extraction is very interesting topic and present research should be interesting to Molecules readers. However, present approach is still lacking in novelty and some important aspects which should be improved prior the further evaluation. My main remarks are:
- State-of-the-art of recent papers focused on UAE of carotenoids should be added in introduction.
- RSM part in introduction should cite case studies focused on optimization of carotenoids extraction, rather than anthocyanins and other examples.
- At the end of introduction, highlight the novelty of this work comparing to previous publications on this topic.
- 6. Add information how were results expressed.
- 7. Add information and explain how did you choose experimental design. Was it literature data, preliminary experiments, etc.?
- 7. How did you select experimental domain for independent variables?
- 8. Please explain how did you determine model fitness using F-test, coefficient of correlation, lack of fit testing and other statistical parameters.
- Why did you selected this method of extraction rather than green technique such as supercritical fluid extraction?
- Please compare your results with literature data on SFE of carotenoids from red pepper.
- Add discussion on significant lack of fit from Table 4. How does that affect model fitness?
- What about rather poor R2 from Table 4 and 6? You should improve your discussion.
- Information about desirability function used for multi response optimization is missing. Add that in 4.8. and 2.4. segments. What was the D function value?
- Improve conclusions with some information about potential application of these results.
- Paper should be proof read by the native English speaker since it has certain grammatical and style errors.
Reviewer 2 Report
This paper deals with the optimisation of ultrasonic extraction of three selected carotenoids from pepper leaves using the RSM method.
Unfortunately, the authors are trying to publish in different form a part of an already previously published paper in 2017 in the Journal of Agricultural and Food Chemistry (https://doi.org/10.1021/acs.jafc.7b02440):
Reversed-Phase High-Performance Liquid Chromatography for the Quantification and Optimization for Extracting 10 Kinds of Carotenoids in Pepper (Capsicum annuum L.) Leaves
Jing Li†, Jianming Xie*†, Jihua Yu†, Jian Lv†, Junfeng Zhang†‡, Xiaolong Wang†, Cheng Wang†, Chaonan Tang†, Yingchun Zhang†, Mohammed Mujitaba Dawuda†§, Daiqiang Zhu†, and Guoli Ma†
Agric. Food Chem. 2017, 65, 38, 8475–8488, Publication Date: August 25, 2017
More specifically, the research methodology and extraction optimization parameters are identical in the reviewed paper to those in the already published one. The authors decided to present them this time in the form of RSM studies. The only difference is that optimisation was performed for 3 of the 10 previously determined carotenoid substances.
Disregarding the fact of self-plagiarism, the reviewed article requires major linguistic corrections.
In the introduction there is no justification why the authors extract carotenoids from leaves, what practical effect the extraction of these substances is supposed to bring.
Methodology of extraction and sample preparation for HPLC analyses as well as instrumental analysis itself are scantly described. No information is given to allow the method to be reproduced in other laboratories. The methodology in several places suggests that such studies have already been performed: indication of the best column temperature, phase composition (but no gradient over time is given). Validation is also described briefly and imprecisely in contrast to that given in the 2017 publication. The number of replicates and LOD, LOQ, recovery values, as well as range of concentrations taken to create standard curves are missing.
The results and discussion are limited, presumably to avoid repeating the description and discussion of optimisation from the 2017 publication.
Conclusions are general, some can be counted as text extracted from the discussion of results. I will disagree with conclusion 3 - extraction is not simple and the authors have not demonstrated that extraction efficiency is high.
In view of the above, I strongly recommend that the article be rejected for publication in the MDPI publisher.